# Organic Heterojunction Devices Based on Phthalocyanines: A New Approach to Gas Chemosensing

**DOI:** 10.3390/s20174700

**Published:** 2020-08-20

**Authors:** Abhishek Kumar, Rita Meunier-Prest, Marcel Bouvet

**Affiliations:** Institut de Chimie Moléculaire de l’Université de Bourgogne (ICMUB), UMR CNRS 6302, Université Bourgogne Franche-Comté, 9 avenue Alain Savary, 21078 Dijon CEDEX, France; maria-rita.meunier-prest@u-bourgogne.fr

**Keywords:** gas sensors, organic heterojunction effects, phthalocyanines, heterostucture, conductometric transducers

## Abstract

Organic heterostructures have emerged as highly promising transducers to realize high performance gas sensors. The key reason for such a huge interest in these devices is the associated organic heterojunction effect in which opposite free charges are accumulated at the interface making it highly conducting, which can be exploited in producing highly sensitive and faster response kinetics gas sensors. Metal phthalocyanines (MPc) have been extensively studied to fabricate organic heterostructures because of the large possibilities of structural engineering which are correlated with their bulk thin film properties. Accordingly, in this review, we have performed a comprehensive literature survey of the recent researches reported about MPc based organic heterostructures and their application in gas sensors. These heterostructures were used in Organic Field-Effect Transistor and Molecular Semiconductor—Doped Insulator sensing device configurations, in which change in their electrical properties such as field-effect mobility and saturation current in the former and current at a fixed bias in the latter under redox gases exposure were assessed to determine the chemosensing performances. These sensing devices have shown very high sensitivity to redox gases like nitrogen dioxide (NO_2_), ozone and ammonia (NH_3_), which monitoring is indispensable for implementing environmental guidelines. Some of these sensors exhibited ultrahigh sensitivity to NH_3_ demonstrated by a detection limit of 140 ppb and excellent signal stability under variable humidity, making them among the best NH_3_ sensors.

## 1. Introduction

Over the recent few decades, organic semiconductors have drawn tremendous attention in the development of electronic devices such as organic light-emitting diodes (OLEDs) [1], organic photovoltaics (OPVs) [2], organic field-effect transistors (OFETs) [3], organic lasers [4], memory devices [5] and chemical sensors [6]. Some of these devices have already attained innovation maturity and have been transferred from the benches of research labs to the shelves of the market. A common example is OLEDs, which is widely used in new generations of flat panel color displays in products like televisions, monitors, smartphones, tablets and hosts of wearable electronics while OPVs are increasingly being recognized as a potential alternative to conventional expensive silicon-based photovoltaics [7]. Chemical sensors are another widely researched area inviting numerous studies on organic semiconductors owing to their advantages of low cost and flexibilities of chemical design, synthesis and processing [8].

Conventionally, the improvement in organic semiconductor-based chemical sensors was attempted by changing the sensing materials, but recently a growing trend aiming to exploit the role of interfaces to modulate the sensing performance has been observed. The interfaces can be created between the sensing material thin film and the electrodes (usually metals), which has invited comprehensive studies on metal-organic junctions and their implications on chemical sensors [9]. In fact, in many sensor device configurations, either metal electrodes or organic semiconductor thin film are exposed to target gases and their interactions induce a modulation of charge carrier injection from metal to organic film and thus interfacial alignment of energy levels at metal-organic junction, which also forms the basis of sensing principle [10,11,12]. In addition to metal-organic junction, an interface can also be created if the sensing device configuration consists of a bilayer or multilayer of sensing active layer, forming an organic-organic heterojunction. A typical p-n heterojunction formed from p-type and n-type organic semiconductors is characterized by the accumulation of electrons (e^−^) and holes (h^+^) free charge carriers at the interface because of the electron affinity and work function differences forming the space charge region (Figure 1). Such interfaces are usually more conducting than the bulk of either semiconductor forming the heterojunction and carrier transport takes place along the interface. On the contrary, space charge region in the conventional inorganic p-n junction are highly resistive because they are composed of depletion of free charge carriers and have opposite polarity of the heterojunction in-built electric field. Since it is well known that bulk carrier transport in organic semiconductor thin films devices are slow owing to high density of traps which restrict the fast hopping of e^−^ or h^+^ [13] therefore, creation of a heterojunction in organic semiconductor-based devices is highly advantageous to enhance the carrier transport and thus device conductivity. The dynamics of interfacial alignments of free charges can be modulated by an external doping such as chemical doping by redox gases (e.g., NO_2_ or NH_3_), which has been exploited to develop high performances gas sensors [14,15,16,17].

For gas chemosensing applications, organic heterojunctions are created in diverse device configurations, such as conventional two-terminal diode, a three-terminal ChemFET, and a bilayer heterojunction, initially called as Molecular Semiconductor - Doped Insulator (MSDI) heterojunction, which was invented and patented by one of us [19,20]. These devices employ a wide family of organic semiconductors comprising conducting polymers and molecular materials. The former are usually characterized by long chains with poorly defined elemental compositions while the latter, as the name implies, have fixed elemental compositions, low molecular weight molecules and inter-molecular forces between two adjacent molecules of less than 10 kcal mol^−1^ [21]. Some of the commonly used conducting polymers are poly-3-hexylthiophene (P3HT), polyaniline (PANI), and polypyrrole, among others, which have found ample gas sensing applications [22,23]. However, sensing applications of these materials are restricted because of their low thermal and chemical stabilities. Molecular materials, such as pentacene, perylene, fullerene, phthalocyanines manifest rather a more versatile chemosensing application owing to their high thermal and chemical stabilities and can be deposited into thin films by solution processing as well as high temperature vacuum sublimation. Among different molecular materials, phthalocyanines are undoubtedly the most studied molecular semiconductors in chemosensing, which is attributed to their high dark conductivity variations upon exposure to redox gases, excellent chemical and thermal stabilities and most importantly, possibility to tune semiconducting properties by molecular engineering such as changing the central metal atom or peripheral substituents. In fact, structure is so strongly correlated with the physical properties that a subtle change in the molecular structure, like substituting a metal or ligand can markedly alter its bulk physical properties like conductivities, band gap, solubility and thermal stability. For example, copper phthalocyanine (CuPc) is a p-type semiconductor in air, while its perfluorinated analogue Cu(F_16_Pc) is one of the few n-type semiconductors stable in air. Change of metal center also has a strong influence on electronic properties, such as lutetium-bis-phthalocyanine has shown unique radical nature and is the first intrinsic molecular semiconductor reported with exceptionally high free charge carriers density (5 × 10^16^ cm^−3^), electronic conductivity (5 × 10^−5^ Ω^−1^ cm^−1^) [21] and very low band gap (0.5 eV). Owing to such fascinating electrical properties and stabilities of MPc thin films, these molecular semiconductors have stimulated a lot of interests in organic heterojunction-based gas sensors.

Accordingly, the purpose of this review is to introduce the reader to contemporary chemosensing transducers incorporating MPc-based heterostructures. In the past, different reviews encompassing MPc materials application in chemical sensors were reported, highlighting the advancements in sensing properties [24,25,26,27], but none of these literature surveys focused specifically on phthalocyanine heterostructures. To the best of our knowledge, only one review by Wang et al. [28] has covered the area of organic heterostructures including MPc, but its area of focused was devoted in understanding the fundamental electronic properties. The present review is highly novel because for the first time a systematic literature survey of MPc heterostructures application in gas sensors in the last 10 years is presented. In this endeavor, a concise discussion on organic heterojunction gas sensors and their metrological parameters evaluation are presented first. The suitability and relevance of MPc materials in fabricating gas sensors devices are highlighted with an emphasis on organic heterojunction gas sensors. Furthermore, organic heterojunction effects, their origin and different types in MPc based heterostructures are comprehensively described with a focus on electronic energy level alignments and charge transport at the interface of the heterojunction. Additionally, the integration of MPc heterostructures in chemosensing devices, in particular different configurations of OFET and MSDI is also discussed. A major focus is then given to an extensive survey of OFET and MSDI gas sensing devices incorporating MPc-based heterostructures and the advantages of exploiting organic heterojunction effects in improving sensors metrological parameters are demonstrated. Finally, a concise conclusion and a perspective regarding the enhancement and efficient utilization of organic heterojunction effects in gas sensing devices are given.

## 2. Suitability of Phthalocyanines in Heterostructure Based Gas Sensors

### 2.1. Working Principles and Metrological Parameters

A simplified scheme of sensing mechanism for gas sensors employing an organic heterostructure and exploiting its electrical properties is presented in Figure 2. According to this, redox gases can either donate or withdraw electrons from the heterostructure sensing layer depending on the oxidizing or reducing nature of the gas molecules. As a result of the chemical doping in the sensing layer, the transducer electrical output changes as a function of gas concentration. For example, under exposure to an electron donating gas such as NH_3_, the e^−^ concentration in n-type heterostructure increases, which results in a current rise in the transducer output circuit and vice versa for p-type. On the other hand, in the presence of an electron acceptor gas like NO_2_, e^−^ in the n-type heterostructure sensing layer decrease, resulting in a decrease in the current of the transducer circuit. However, such a simplified mechanism can change, depending on the architecture of the heterostructure and its integration within the transducer circuit. Usually, heterostructures are deposited in the form of a bilayer or a homogeneous blend of two dissimilar organic semiconductors. Thus, there can be either one organic-organic junction (bilayer film) or an interpenetrating network of junctions in whole heterostructure volume (blend film). The interest in organic heterostructures for gas sensing application is driven by the improvement in the sensors’ metrological and analytical parameters, which are evaluated by static and dynamic operations of the sensors. A brief survey of these parameters is presented here to further demonstrate their improvements in the later sections as the advantages of using heterostructures in gas sensors.

The key metrological characteristics of gas sensors can be listed as sensitivity, selectivity, linear operational range, response and recovery time, detection limit, stability and reproducibility. It should be noted though, taking into account a huge diversity in sensors types and their working environment, that it is difficult to give a uniform definition of these analytical parameters, holding true for all types of sensors, so a definition more adapted for a gas sensor is presented. Sensitivity is defined as the change in output signal of the sensor per unit change in the concentration of input physical analyte (gas being measured). Operatively, it is estimated by the slope of sensors output characteristic curve (ΔY/ΔX; ΔY represents the change in sensor output signal associated with ΔX input analyte). In electrical gas sensors, output signal is often expressed by relative response (RR) (ΔI/I_0_; ΔI is the change in current from the initial current I_0_ upon gas exposure). Accordingly sensitivity refers to the change in RR per unit change in the exposed gas concentration (usually in ppm), which is quantitatively estimated by the slope of the RR versus gas concentration curve. Selectivity represents the ability of a sensor to produce an output signal corresponding to a specific input analyte at a given experimental condition when exposed to a mixture of input species. In gas sensors, selectivity is commonly assessed by exposing the sensors to a mixture of gases and comparing the obtained sensor response with the one realized in the presence of only the targeted gas. Dynamic range of a gas sensor is particularly important in application areas where target gas concentration fluctuates in a wide concentration range and represents the minimum and maximum values of the input (gas concentrations) that can be precisely measured. Limit of detection (LOD) is another sensor’s analytical parameter, which refers to the input analyte concentration associated with the minimum output signal that can be measured with reasonable accuracy for the selected analytical method. Operatively, different approaches are used to estimate the LOD of gas sensors, which depend on the associated signal transduction method. As reported in the tutorial review on OFET based gas sensors by Torsi et al. [29], LOD represents the gas concentration associated with device response 3-times larger than standard deviation of its blank signal (sensors response in the absence of gas and also corresponds to baseline), which is deduced from the calibration curve. It is often denoted as (3 × N/S), where N is the noise of the blank signal and S is the sensitivity of sensors. Majority of gas sensors studies adopt this equation to determine LOD. Kinetics of sensors responses, i.e., response and recovery speeds, are other key parameters used to assess the metrological performance of a gas sensor, which are quantified by response and recovery times. The response time is the time taken by a sensor to change its output signal upon input analyte exposure (subjected to a constant gas concentration) from its initial state to a final steady state value within a tolerance band of the correct new value. For example, response time (90%) or t_90%_, which is also extensively inferred in many gas sensing studies, signifies the time required for a sensor output signal to change from its baseline value to 90% of the steady state value when sensor is under gas exposure. The characteristic time corresponding to return to the initial state (stimulus or gas removed) is called recovery time. More details about gas sensors analytical parameters can be found elsewhere in the literature specifically dedicated to these topics [30].

### 2.2. Metal Phthalocyanines in Organic Heterostructures

MPcs are some of the typical molecular semiconductors in organic electronics, drawing tremendous interest owing to their relatively easy availability, either commercially or through established synthesis methods and their stable properties in ambient environments. The molecular unit consists of a planar macrocycle constituted by four isoindole groups connected through azamethine bridges (Figure 3) and a centrally located metal atom. Such a macrocyclic arrangement imparts a π-conjugated structure to the molecule and results in diverse electronic, optical and semiconducting properties, depending on the type of metal atom present. The effect of the metal center in the phthalocyanine on its electronic properties has been extensively studied, theoretically as well as experimentally, reporting a strong correlation of these properties with the nature of metal atom [31,32,33]. According to these studies performed on main group transition metal phthalocyanines, frontier orbitals (Highest Occupied Molecular Orbital (HOMO) and Lowest Unoccupied Molecular Orbital (LUMO)) of MnPc and FePc have a major contribution from metal 3d-orbitals, CoPc has lesser influence from Co 3d-orbitals while NiPc, CuPc and ZnPc have negligible effects of metal 3d-orbitals. Accordingly, electron affinity, workfunction and optical absorption change as a function of metal center [34]. The metal atom in MPc also determines its planarity and thus its electrical properties. The main group transition metal atoms fit into the cyclic core and form a planar structure, but lanthanide group metals such as Lu, Eu, Gd are too large to fit into the central core, thus forming a double decker or triple decker phthalocyanine [35]. These multi-decker phthalocyanines are highly conducting because of their radical nature and have a narrow energy gap between frontier molecular orbitals [21]. These properties are further modulated by substitution of peripheral hydrogens or grafting at the metal center in MPc with an electron donating or withdrawing group, which has been systematically reviewed by Nyokong et al. [36]. The application of MPc in gas sensors can be traced back to late 1980s in the pioneering work of Jones et al., reporting the very high responses and ultrafast kinetics (a few s) of PbPc-based chemiresistors towards NO_2_ obtained by thermal cycling at 300 °C [37]. In the subsequent years, extensive reviews on OFET devices utilizing MPc materials were reported [8,38] including one by us which focused in particular on gas sensor applications [39]. MPc-based chemiresistors were extensively studied by Kummel and coworkers to detect a large group of electron-donating gas analytes [40]. Sensors responses were correlated with the Lewis basicity and hydrogen binding strength of gas molecules, which depended on the central metal atom in MPc (M: Co, Ni, Cu, Zn). The sensitivity and the response/recovery kinetics of CoPc were better than other MPc and H_2_Pc owing to the stronger interaction of Co with the gas molecules, which is also attributed to the higher electron affinity of CoPc. The sensors responses were also linked with the thermodynamic parameters and an exponential correlation with binding enthalpy of gas molecule with metal center in MPc was proposed in which CoPc has the highest while H_2_Pc has the lowest binding enthalpies. Finally, to achieve discrimination in sensors response, principal component analysis was applied. In another work from the same group, the effect of metal centers in MPc was studied to detect H_2_O_2_ vapors and CoPc exhibited a decrease in current while NiPc, CuPc and H_2_Pc showed an opposite trend upon gas exposure [41].

The motivation behind using a MPc heterostructure as a sensing layer is to overcome the metrological parameter limitations observed when using only one MPc. For example, chemiresitors based on polycrystalline MPc present very high sensitivity towards oxidizing gases like NO_2_, but suffer from a prolonged recovery. Such a slow recovery was attributed to strong chemisorption of the analyte gases on the sensor surface [42], diffusion of the gas molecules in the bulk film and highly discrete pathways for the charge transport [13]. To speed up the response and recovery kinetics, different approaches have been adopted, such as sensing measurements at elevated temperature [43], use of ultrathin layer of MPc films [44] and in some cases, application of a single crystal MPc as a sensing film [45]. Despite overcoming the problem of the slow response kinetics of sensors, these methods are complex and expensive and cannot be applied to large scale monitoring. Moreover, continuous exposure of MPc film in the ambient air can cause irreversible adsorption of H_2_O and O_2_, which can negatively impact the gas sensing properties, particularly for ultra-thin sensing layers and n-type MPcs, because these adsorbents act as traps and decrease the charge mobility in the film [46]. A simplified approach to fasten the sensors response-recovery kinetics is to use the MPc-based heterostructure as an active sensing layer. Indeed, the numerous studies on Cu(F_16_Pc)/LuPc_2_ heterostructure-based conductometric sensors by Bouvet and coworkers demonstrated faster kinetics, stable response in variable humidity, higher sensitivity and lower LOD for NH_3_ detection as compared to the device utilizing only LuPc_2_ as a sensing layer [14,47].

The high relevance of MPc in the fabrication of heterostructures for electrical gas sensors applications are attributed to the strong intermolecular overlapping of frontier orbitals of MPcs associated in the heterostructure formation, resulting in the faster mobility of charge carriers. Such overlapping can be further optimized by modulating the orbital energies through attaching appropriate electron donor or acceptor moieties on the macrocyclic periphery. Moreover, the workfunction of MPcs can also be tuned by the electronic effects of substituents, such that electron donating groups decrease while electron accepting groups increase the MPc workfunction. This can be understood as electron-donating substituents increasing the electron density in the macrocycle, which lifts its Fermi energy (E_F_) towards vacuum level while electron-withdrawing groups extract electron density from the macrocycle, pushing E_F_ away from vacuum level. For example, substitution by an electron-accepting group such as fluorine increases the workfunction of phthalocyanines because of a lowering of HOMO and LUMO energies [48,49]. Therefore, the workfunction of MPc can be finely engineered to align with the electrode (such as gold or ITO) workfunctions for an efficient charge injection and reception during sensors operation. Indeed, the workfunctions of some of the commonly used electroactive MPcs (LuPc_2_, CuPc, CoPc, Cu(F_16_Pc)) match the Au or ITO workfunctions. Moreover, in a bilayer heterostructure of two MPcs, the dense molecular packing of top layer protects the layer below from irreversible chemisorption of O_2_ and H_2_O, which is particularly important to maintain n-type or ambipolar sensing properties of MPc heterostructures [50,51] and from oxidation by strong oxidizing gases (ozone and NO_2_) [52]. Above all, however, the main advantage of using MPc heterostructures instead of a homogeneous MPc film is to benefit from the organic heterojunction effects in which free charge carriers are accumulated at the junction.

## 3. Organic Heterojunction Effects and Chemosensing Devices

### 3.1. Interfacial Energy Levels Alignment and Charge Distribution

Organic heterojunction effects have been observed in different chemosensing devices employing MPc-based heterostructures as an active layer. The origin of organic heterojunction effects lie at the interfacial alignments of electronic energy levels because of the workfunction differences (ΔΦ) between the organic semiconductors in the heterostructures. Ultimately ΔΦ determines the direction of charge carriers’ transport and then the subsequent interfacial charge redistribution at the interface. A scheme of interfacial charge transfer between energy levels in CuPc/LuPc_2_ and Cu(F_16_Pc)/LuPc_2_ bilayer heterostructures is shown in Figure 4a, which is adapted from our recent works on chemosensing devices employing these heterostructures [47,53]. For a CuPc/LuPc_2_ bilayer heterostructure, both the constituents are p-type molecular semiconductors and Φ_LuPc2_ > Φ_CuPc_ [48,54], accordingly the e^−^ transfer takes place from the HOMO of CuPc to the semi-occupied molecular orbital (SOMO) of LuPc_2_ and h^+^ transport follows the opposite direction through path-1 in order to equilibrate the E_F_ at the interface. On the other hand, in Cu(F_16_Pc)/LuPc_2_ heterostructures having a combination of n- and p-type molecular semiconductors, because of Φ_LuPc2_ < Φ_Cu(F16Pc)_, electron transfer occurs from the SOMO of LuPc_2_ to the LUMO of Cu(F_16_Pc) through path-2 (Figure 4a).

As a consequence of e^−^ and h^+^ hopping between the frontier orbitals of MPcs, the charges are redistributed at the interface. Because of the h^+^ transfer from LuPc_2_ to CuPc in p-p isotype heterostructure, h^+^ is depleted in LuPc_2_ and accumulated in CuPc near the interface, forming a depletion/accumulation heterojunction (Figure 4b). The h^+^ injection in CuPc causes upward HOMO band bending while h^+^ depletion in LuPc_2_ results in downward SOMO band bending (Figure 4d). In Cu(F_16_Pc)/LuPc_2_ n-p anisotype heterostructures, because of the e^-^ injection in the Cu(F_16_Pc) layer, which also creates an equal h^+^ injection in LuPc_2_ layer, e^−^ and h^+^ are accumulated near the interface in the Cu(F_16_Pc) and LuPc_2_ layers, respectively (Figure 4c) and such a heterojunction is commonly known as accumulation type. In this heterojunction, the LUMO level of Cu(F_16_Pc) and SOMO level of LuPc_2_ are bent downward and upward, respectively, for efficient charge transfer across the interface (Figure 4e). The accumulation heterojunctions are highly conducting, because of the filling of trap states, which make the interfacial charges highly mobile, which also accelerates the kinetics of the chemosensor response. A comprehensive survey of different types of organic heterojunctions has been made previously [18,28,55], among those accumulation and accumulation/depletion heterojunctions are most commonly observed in MPc based heterostructures.

The formation of the heterojunction and e^−^ and h^+^ charges accumulations at the interface were validated by an ultraviolet photoelectron spectroscopy (UPS) study of the Cu(F_16_Pc)/CuPc heterostructure showing the creation of an interface dipole and an apparent band bending on both sides of the heterojunction [56]. The HOMO and LUMO of CuPc are bent upward (towards the higher binding energy) while those of Cu(F_16_Pc) are bent downward (towards the lower binding energy) within a range of 15 nm from the interface on each side of the heterojunction (Figure 5). Moreover, transfer of electrons from the CuPc side to Cu(F_16_Pc) creating an accumulation of e^−^ and h^+^ at the interface was also demonstrated. The free carrier’s density in the space charge region was estimated to be about 10^18^ cm^−3^, which was six orders of magnitude higher than the bulk carrier’s density. Moreover, such organic heterojunction effects were observed in bilayer and homogenous blend of Cu(F_16_Pc)/CuPc heterostructures incorporated in different device configurations (OFET and diode) [57,58].

### 3.2. MPc Heterostructure Integration in Chemosensing Devices

There are mainly two device structures studied so far, incorporating MPc-based heterostructures for chemosensing application; which are OFET and MSDI. The former offers a much broader range of device configuration designs, as highlighted in recent reviews devoted to OFET-based gas sensors [12,59,60]. Among different OFET designs employing organic heterostructures, the two most commonly used ones are the suspended-gate and top-gate configurations shown in Figure 6a,b, which are distinguished by the relative position of dielectric and gate components in the device. In the former design, an organic semiconducting layer is exposed to the target gas analyte while in the latter a gate is exposed to the target gas. The choice of either scheme depends on the nature of the interaction between the exposed gases and the gates or semiconducting layers. A unique characteristic of OFET devices employing a bilayer of two organic semiconductors is that a conduction channel exists at the interface of two organic semiconductors contrary to the conventional OFETs in which a conduction channel exists at the dielectric/semiconductor interface [28]. This is because of the organic heterojunction effects in which opposite charges (e^−^ and h^+^) are accumulated at the interface of two organic semiconductor having different workfunctions. Such an interfacial charge redistribution creates a space charge region at the organic-organic junction. In OFET devices having MPc-based heterostructures as active layer, the strength of the space charge region is characterized by estimating the carrier mobility at the organic-organic junction. For example, the formation of an accumulation heterojunction was noted in Cu(F_16_Pc)/CuPc heterostructure-based organic field-effect transistor (OFET) by Wang and coworkers [28,57]. In these comparative works, the heterostructure-based OFET remained in on-state (a conduction channel exists even at zero gate bias) with 1.2 µA source to drain current, which is equal to the current of a CuPc-based OFET at −40 V gate bias. The device exhibited h^+^ accumulation and depletion modes, respectively, on increasing and decreasing the gate voltage. In a similar heterostructure studied by Wei, the OFET exhibited air stable ambipolar (both e^-^ and h^+^ conduction channel) carrier transport, with e^-^ and h^+^ mobility at 20 °C as 8.69 × 10^−3^ and 1.40 × 10^−2^ cm^2^ V^−1^ s^−1^ [61].

The MSDI is an original device invented by one of us [19], which is characterized by the arrangement of a semiconducting bilayer on interdigitated electrodes (Figure 6c), such that the top layer has a very high carrier concentration while the sublayer is a relatively poor conductor. Under thermodynamically suitable conditions, e^-^ or h^+^ are injected in the sublayer, which also justify its name as molecular semiconductor - doped insulator. One should not assume that saying insulator indicates the sublayer is non-conducting, but rather that its carrier density is comparatively much lower than in the top layer, so it is named symbolically like that.

The main reason for adopting such a device configuration is to benefit from the organic heterojunction effects, producing high mobility of free charge carriers at the interface. Because of the poor conductivity of the sublayer, the electronic injection from the electrode follows a path through the highly conducting interface. Thus, charge transport in MSDI devices mainly takes place along the interface in the sublayer. Accordingly, the response of the sensor under gas exposure is determined by the semiconducting nature of the sublayer. A common example of an organic heterostructure satisfying this condition is Cu(F_16_Pc)/LuPc_2_, which has been extensively investigated by us for the development of chemosensors applied for redox gas detection [20,47,53,62] exhibiting a current increase in the presence of electron-donating gases and a current decrease in the presence of electron accepting gases, in accordance with the n-type nature of the sublayer.

Another MPc-based heterostructure design recently studied by us in chemosensing applications was termed double lateral heterojunction (Figure 6d) in which each strand of the interdigitated electrode was electrochemically coated by a poor conducting polymer such that the gap between two neighboring electrode strands remains uncoated, followed by a homogeneous coating of a highly conducting top layer [15,63]. In this configuration, charge transport takes place laterally owing to the presence of a conducting zone between the strands of the coated interdigitated electrodes.

The variation of space charge region or energy barrier in MSDI or double lateral heterojunction devices was extensively studied by Mateos et al. [14,63]. In these works, the strength of the organic heterojunction has been modulated by electrografting substituted benzenes on the ITO electrode surface, which is further used to fabricate a bilayer organic heterostructures with Cu(F_16_Pc) and LuPc_2_ (Figure 7b). The apparent interfacial energy barrier (U_th_), equivalent to the x-intercept of tangent to the current-voltage (I(V)) curve at the maximum bias (Figure 7a) was enhanced by grafting a substituted benzene on the ITO. The maximum increase in U_th_ was obtained for a tetrafluorobenzene (TFBz)-grafted heterostructure while a minimum enhancement was noted for a dimethoxybenzene (DMBz)-grafted device. Benzene (Bz) and trifluoroethoxybenezene (TFEBz)-grafted devices showed intermediate U_th_ values (Figure 7c). Such interfacial energy barrier tuning also improved the NH_3_ sensing performance of the conductometric transducers, as demonstrated by a sub-ppm detection limit (140 ppb), higher sensitivity and negligible interference from relative humidity (rh) fluctuation [14,15,63].

## 4. Chemosensing Properties of MPc-Based Heterostructures

### 4.1. OFET Based Gas Sensors

Although, OFET devices employing MPc heterostructures have been widely studied, their applications in gas chemosensing are rather scarce despite plenty of reviews of OFET gas sensors based on homostructures [12,59,64,65]. The majority of reported research on MPc heterostructure-based OFET is focused on fundamental electrical transport property studies and other organic electronics applications. Nonetheless, a few MPc heterostructure-based OFET devices have drawn interest in gas sensor development in the last 10 years, which are being reviewed hereinafter. The chemosensing properties of a Cu(F_16_Pc)/CuPc bilayer-based OFET were studied by Zhang et al. for NO_2_ detection [66]. The OFET device structure consisted of a suspended gate configuration (Figure 8a) and was fabricated using a heavily doped n-type monocrystalline Si as substrate, SiO_2_ as the dielectric layer and gold-titanium as source and drain electrodes, respectively. The conduction channel existed at the interface of CuPc and Cu(F_16_Pc) as an interpenetrating network in each MPc domain. Upon exposure to 20 ppm of NO_2_, source to drain current decreases (Figure 8b), revealing n-type behavior of the OFET and e^−^ accumulation in Cu(F_16_Pc), forming the device conduction channel. The sensing properties of the device were further optimized by changing the thickness of CuPc layer and the appropriate combination was 15 nm of CuPc and 40 nm of Cu(F_16_Pc) to produce the maximum RR. In fact, a larger thickness of the top-layer prevented the NO_2_ molecules reaching the highly conducting interface, thereby decreasing the RR. Another OFET sensor, highly sensitive for NO_2_, was realized by using a CuPc/pentacene heterostructure in a top gate device configuration (Figure 8c) and incorporating an ITO substrate and zinc oxide/poly(methyl methacrylate) (ZnO/PMMA) as dielectric [67]. The role of ZnO nanoparticles was highlighted as they transformed the conduction channel in the OFET device from an organic/organic interface to the combination of dielectric/organic and organic-organic interfaces. The RR values estimated from OFET characteristics such as saturation current (I_on_) and field-effect mobility (µ) experience an increase of 193% and 69%, respectively, under exposure to 15 ppm of NO_2_ in the presence of ZnO while a decrease of 30% and 20%, respectively, in the absence of ZnO. Thus, ZnO has a synergistic effect on NO_2_ response concomitantly reversing the device polarity to p-type. The sensor response was easily distinguishable for different NO_2_ concentrations under 10 min exposure-recovery steps (Figure 8d) exhibiting 9% RR for 0.5 ppm of NO_2_ and its response was affected neither by 50% of rh nor by SO_2_. Moreover, response of the device was very stable because after storing in ambient environment for 30 days, no significant decline in its metrological performances was noted.

Recently, Fan et al., reported an OFET device for NO_2_ detection based on p-type CuPc and n-type dioctylperylene tetracarboxylic diimide (PTCDI-C8) heterostructure on ITO substrate and PMMA dielectric layer (Figure 9a) [68]. The chemosensing properties of the device were optimized by depositing 7 nm of CuPc and different thicknesses of PTCDI-C8 over it, out of which the device with 0.5 nm exhibited the highest sensitivity to NO_2,_ although higher field-effect mobility obtained with 2 nm PTCDI-C8. The transfer curve of OFET shows a p-type behavior indicating the presence of a conduction channel in the CuPc layer at the interface. The higher sensitivity of the OFET with thinner PTCDI layer was attributed to the easier interaction of NO_2_ with the conduction channel. Transfer curves of the device (Figure 9b) experienced an increase in the saturation current upon NO_2_ exposure from 2 to 30 ppm. The RR values assessed from the change in saturation current were obtained as 45% and 126% at 2 ppm and 30 ppm of NO_2_ exposure, respectively, which was 6-times larger than RR obtained by using only a CuPc layer (Figure 9c), demonstrating the advantages of heterojunction effects. The CuPc/PTCDI-C8 heterostructure revealed 10- fold higher sensitivity than a CuPc based OFET towards NO_2_.

Heterostructures having a homogeneous blend of CuPc or CoPc with tris(pentafluorophenyl) borane (TPFB) (prepared by co-evaporation) were studied in OFET device design (Figure 9d) for detection of NH_3_ vapor [69]. Transfer characteristics of the OFET device although experience decrease in drain current and field-effect mobility after the addition of TPFB, which is a strong electron acceptor, but NH_3_ sensing properties were improved.

Upon NH_3_ exposure, the drain current decreases, highlighting the p-type conduction channel of the device. The RR estimated from the % decrease in the drain current was obtained as 33% and 37% under 4.5 ppm and 12% and 13% under 0.45 ppm of NH_3_ for CuPc/TPFB and CoPc/TPFB, respectively, which is much larger than the responses obtained without TPFB. Such a high sensitivity allowed achieving a LOD value down to 350 ppb. The beneficial role of TPFB was attributed to its strong electron-accepting nature, because of that it acts as h^+^ trap center and also interacts with NH_3_ molecules through hydrogen bonding facilitating facile e^-^ injection from NH_3_ to the semiconductor layer. Such electronic effects of this molecule were confirmed because, by using less electron accepting molecules like triphenylmethane (TPM) or triphenylborane (TFB) as additives in the heterostructure, a lower RR was obtained (Figure 9e). The selectivity of the device was evaluated by comparing RR with eight different interfering gases. It is evident from Figure 9f, RR value is largest for NH_3_ (9 in the x-axis), but notable interferences from isopropylamine (6), isobutylamine (7) and H_2_S (8) were observed. The sensor exhibited very high long-term stability in its response towards NH_3_ because similar current change was noticed by storing the devices in a sealed container maintained at −30 °C.

Ji et al., exploited a double heterojunction effect in an ultrathin trilayer heterostructure of *para*-hexaphenyl (p-6P), N,N′-diphenylperylene tetracarboxylic diimide (PTCDI-Ph) and vanadyl phthalocyanine (VOPc) in an OFET device configuration having a 3 mm wide sensing area and extended electrodes (Figure 10a) for a highly sensitive detection of NO_2_ [44].

The advantage of double heterojunction effects was exemplified as the sensor calibration curve (RR vs NO_2_ concentration) indicates a much higher slope for double heterojunction OFET from single heterojunction OFET (Figure 10b) highlighting the higher sensitivity in the former. The RR of the double heterojunction OFET sensor under alternate exposure-recovery cycles of NO_2_ (concentration range 5–30 ppm) has been shown in Figure 10c which is approximately 4-times higher than the OFET device having one heterojunction between p-6P and PTCDI-Ph. However, RR at 30 ppm of NO_2_ of double heterojunction device experienced a 42% decrease after storing over one month under ambient conditions (room temperature and atmospheric air).

A highly sensitive NO_2_ sensor based on ultrathin OFET device incorporating a 1.8 nm bilayer of titanyl phthalocyanine (TiOPc) film as sublayer and 1 nm Cu(F_16_Pc) film as top layer was reported by Wang et al. [70]. It is evident from Figure 10d that the RR of the bilayer OFET sensors towards three different NO_2_ concentrations (2, 3 and 4 ppm) was significantly improved from the similar devices having either of the phthalocyanines. The bilayer OFET device presented a very high sensitivity also demonstrated by sub-ppm detection limit (250 ppb). Such a high sensitivity of the bilayer sensing device was attributed to the organic heterojunction effects in which e^-^ is accumulated at the top Cu(F_16_Pc) film, because of that sensor surface becomes highly reductive for an electron accepting gas NO_2_. However, the bilayer OFET sensor response did not attain a steady state condition during a 30 min NO_2_ exposure and 150 min recovery under clean air as shown in Figure 10e, which requires improvement. Elsewhere Chen and coworkers reported an ambipolar OFET device incorporating solution processed bilayer based on substituted copper phthalocyanines, which exhibited high sensitivity and selectivity towards ethanol [17]. The above reports on redox gas sensing performances of OFET devices highlight that sensing properties such as RR, sensitivity and detection limit are significantly improved when MPc heterostructures are used instead of only one MPc. The long-term stability and reproducibility of these devices are better than conventional OFET sensors. However, MPc heterostructure-based OFET gas sensors are largely unexplored and still remain at the nascent stage of development. Moreover, some of the reported literature works lack extensive investigation of sensing properties such as hysteresis, repeatability, ageing, detection limit, sensitivity, selectivity and linear operational range, which limits a comprehensive evaluation of sensing properties and their comparisons with conventional OFET sensors. It has to be also noted that the sensing properties of these OFET sensors have been studied only for a few gases (mainly NO_2_) and other gases which monitoring are pertinent for air quality control should be investigated.

### 4.2. MSDI Based Gas Sensors

Organic heterojunction effects have been extensively exploited in MSDI devices for gas chemosensing to develop detection platform for oxidizing and reducing gases. One of us (Bouvet et al.) reported the first MSDI device based on CuPc or Cu(F_16_Pc) as a low conducting sublayer and LuPc_2_ as a high conducting top layer [20,52]. Interestingly the CuPc/LuPc_2_ MSDI, experienced a current increase under 90 ppb ozone exposure (electron-acceptor gas) and a current decrease when submitted to 35 ppm NH_3_ (electron-donating gas), displaying the p-type nature of the device (Figure 11a). On the contrary, Cu(F_16_Pc)/LuPc_2_ MSDI revealed opposite trend such that current decrease under ozone and increase under NH_3_ exposure, manifesting the n-type nature of the device. Such variations in the two devices’ response as a function of fluorination of the sublayer phthalocyanine were attributed to the different nature of charge accumulation at the heterojunction interface, as discussed in Figure 4. Thus, apparently, the semiconducting polarity of the sublayer determines the MSDI electrical behavior and its response towards redox gases. These sensors presented very high stability in ambient environments and experience negligible ageing under extended operations and storage in ambient environments. The NH_3_ sensing properties of Cu(F_16_Pc)/LuPc_2_ MSDI were further studied in depth at different rh in a range of 30–70% in light and under dark (Figure 11b) [62,71]. The sensor revealed a stable baseline, high response and small interference from rh. Moreover, the sensors’ responses towards NH_3_ and rh were completely discriminated by applying principal component analysis (PCA).

The chemosensing properties of Cu(F_16_Pc)/LuPc_2_ MSDI were also investigated impedimetrically [47]. Owing to a heterojunction interface, the impedance spectra of the MSDI revealed two semicircles in the Nyquist plot, one associated with bulk resistance (at higher frequency) while a smaller one (at lower frequency) is associated with interfacial charge transfer process (Figure 11c). Different impedance circuit parameters such charge transfer coefficient (α), capacitance (Q_b_) and bulk resistance (R_b_), change upon NH_3_ exposure because of electron donation from NH_3_ disturbs the charges accumulation equilibrium at the heterojunction interface. Accordingly, α and R_b_ decrease as electron donation from NH_3_ makes the heterojunction interface more conducting while Q_b_ increases as shown in Figure 11d. The variations in these parameters were found to be proportional to NH_3_ concentration.

The NH_3_ sensing properties of MSDI having a double-decker MPc as a sublayer as well as top layer were recently reported [72]. Low conducting alkylthio-tetrasubstituted μ-nitridodiiron phthalocyanines were used as sublayer in which the presence of a –2 formal charge in each macrocycle prevented intermolecular e^-^ transfer, while a highly conducting radical LuPc_2_ was used as a top layer (Figure 12a). The MSDI current decreases under NH_3_ exposure, indicating the p-type polarity. The sensor response towards NH_3_ in a concentration range of 10–90 ppm was distinguishable at different concentrations (Figure 12b) and these responses experienced negligible interference from rh variation in a range of 30–60%.

In majority of the MSDI devices reported so far, LuPc_2_ is used as a top layer because of its very high conductivity. Chen et al., reported the use of triple-decker europium complex of phthalocyanine as a top layer, associated with substituted CuPc as sublayer (Figure 12c) [73]. The high solubility of both components of the heterostructure in organic solvents imparted additional advantage of device development by solution processing. The MSDI devices exhibited current decrease under NH_3_ exposure in a range of 50–200 ppm, which strongly depended on the peripheral substitution in sublayer CuPc (Figure 12d) and processing method. The solution-processed MSDI displayed lower response with a high baseline drift while MSDI prepared with vacuum-deposited MPc exhibited higher response. Low response of solution processed MSDI was attributed to the ill-defined and discontinuous interface formation at the heterojunction.

Recently, NH_3_ sensing properties of MSDI fabricated by vacuum sublimated octachloro-complexes of MPc (Figure 13a) as sublayers and LuPc_2_ as top layer were reported by Ouedraogo et al. (including us) [53]. The device current under exposure to 90 ppm of NH_3_ decreases for Zn(Cl_8_Pc) and Cu(Cl_8_Pc), while increases for Co(Cl_8_Pc) revealing p-type polarity for former two and n-type polarity of the latter device (Figure 13b–d). Such variations in the gas sensing properties dependent on the nature of metal center in sublayer MPc were correlated with the higher electron affinity of Co(Cl_8_Pc) resulting in formation of an accumulation heterojunction while the other two MSDI contained h^+^ accumulation/depletion heterojunction. Interestingly, the Zn(Cl_8_Pc)-based device displayed ambipolar behavior after exposure to high rh and NH_3_ concentrations such that device polarity towards NH_3_ is reversed. Such ambipolar behavior was also observed in similar experimental conditions for MSDI having Cu(F_8_Pc) as sublayer (Figure 13f) [74]. The origin of ambipolarity in these devices was attributed to the slow diffusion of NH_3_ and H_2_O in the sublayer after extended exposure, which act as a chemical dopant of e^-^. MSDI devices were highly sensitive to NH_3_ exhibiting stable and discriminated response (Figure 13e) in a range of 10–90 ppm, among which Co(Cl_8_Pc) based device showed sensitivity of 1.48%·ppm^−1^ and LOD of 250 ppb (Figure 13g).

The effects of peripheral substitution with electron donating and accepting groups in sublayer MPc of MSDI device on its NH_3_ sensing properties at different rh was investigated by Wannebroucq et al. [74,75]. Based on the HOMO and LUMO levels determination of different MPc by electrochemical methods, it was observed that the phthalocyanine bearing four alkoxy groups and twelve fluorine atoms behaves approximately as those with eight fluorine atoms. It implies that the electron-donating effect of one alkoxy group compensates the electro-withdrawing effect of one fluorine atom. These sensors based on these MPc sublayer, operated like p-type under ammonia exposure.

Besides using two different MPc inthe MSDI heterostructure, some studies were also made incorporating organic semiconductor other than MPc in the sublayer of MSDI device. One of such works reported preparation of p-MSDI and n-MSDI based on sexithiophene and a perylene diimide derivative (PTCDI) (Figure 14a) as sublayer, respectively, and LuPc_2_ as top layer [76]. The p-MSDI showed current increase under exposure to ozone (400 ppb) and current decrease under NH_3_ (200 ppm) (Figure 14b), while an opposite trend was observed for n-MSDI. n-MSDI based on a similar perylene derivative (perylenetetracarboxylic dianhydride; PTCDA), exhibited high sensitivity to ammonia as depicted in Figure 14c [77]. The response towards three different concentrations of NH_3_ (10, 20 and 30 ppm) was distinguishable, although, changes in rh from 70 to 30% interfered with sensor response towards NH_3_ and the baseline. The calibration curve of the n-MSDI sensor followed Langmuir type growth (Figure 14d) highlighted by saturation of RR at higher NH_3_ concentration indicating the adsorption kinetics playing an important role at higher NH_3_ concentration.

In a similar study, a MSDI based on a triphenodioxazine (TPDO) sublayer was studied for NH_3_ sensing in a wide range of rh [78]. The device presented n-type behavior as highlighted by the current increase under exposure to 90 ppm of NH_3_ and decrease in the recovery step in clean air (Figure 14e). Notably, the baseline of the sensor is very stable over 15 min exposure and one h recovery cycles. The sensor response at short exposure and recovery cycle (1 and 4 min respectively) are shown in Figure 14f. The humidity decrease causes a slight drift of the baseline as well as response associated with NH_3_ exposure, by 4.5% from 70% rh to 10% rh, but this variation is lower than the response to 30 ppm NH_3_. An increase in the RR for NH_3_ was also noted with increasing rh, from 9.1% to 14.8% at 30 ppm NH_3_ when the rh increases from 10% to 70%. Elsewhere, a MSDI comprising an inorganic sublayer tungsten oxide (WO_3_) associated with LuPc_2_ was studied for NH_3_ sensing at different rh [16]. The device presented n-type polarity in accordance with n-type WO_3_ sublayer, high sensitivity (LOD: 250 ppb) and stable response at variable rh between 10–70%.

Very recently, a new MSDI design was investigated by Mateos et al. from our team in which the number of organic-organic junctions was increased by means of electrochemical grafting or electropolymerization of low conducting substituted benzenes on the electrode surface. The objective was to further maximize the organic heterojunction effects in MSDI by creating an additional interface and exploiting it for improvement in NH_3_ sensing performance. In one of the device designs, aniline (ANI), tetrafluoroaniline (TFANI) and dimethoxyaniline (DMA) were electropolymerized on the ITO surface to deposit PANI, PTFANI and PDMA, respectively, and a high conducting LuPc_2_ was vacuum deposited over it [15,63]. Because of the formation of two identical organic-organic junctions, such devices were named double-lateral heterojunctions (DLH, Figure 15a) All three devices displayed p-type semiconducting behavior. Among them PTFANI displayed the best sensing performance towards NH_3_ (Figure 15b). However, it can be noted that discrimination in response at different NH_3_ concentration remains low. Variation of RR with NH_3_ concentration exhibited Langmuir type calibration curves with a saturation in RR values at higher NH_3_ concentration (Figure 15c). Nevertheless, the calibration curve was linear in 1–9 ppm range of NH_3_ concentration and based on that a LOD value of 450 ppb was obtained.

In another design, electrochemical grafting of substituted benzene was realized on an ITO electrode followed by sequential vacuum deposition of Cu(F_16_Pc) and LuPc_2_ [14]. The formation of a low conducting organic film on both electrodes creates additional interface with the sublayer in electromodified MSDI (Figure 15d). Such electrografting greatly improved the NH_3_ sensing performance of MSDI devices from the non-grafted one. The best NH_3_ sensing performance was obtained by the electrografting of DMBz molecule, starting from DMA precursor. The sensors response at different NH_3_ concentrations (10–90 ppm) and rh values are shown in Figure 15e. It can be noted that there is very good discrimination in response at different NH_3_ concentrations. Moreover, rh change has no significant interference in the sensor baseline and its RR values. The calibration curve (Figure 15f) depicting the RR variation with NH_3_ concentration revealed Langmuir type adsorption with a saturation in RR at higher NH_3_ concentration. The LOD value estimated from the linear part of the calibration curve in a range of 1–9 ppm was estimated as 140 ppb, which is among the lowest value reported for NH_3_ sensor. The gas sensing properties of different MPc-based organic heterostructures in OFET and MSDI device configurations are compared in Table 1.

## 5. Conclusions and Outlook

In summary, we have demonstrated that chemosensing devices incorporating MPc-based organic heterostructures have drawn significant research interest for redox gases detection over the past 10 years. The molecular engineering flexibility of the MPc structure, such as changes of metal atoms and substituents, which are also strongly correlated with their bulk electrical properties, make them an ideal material to develop heterostructures with tunable electrical characteristics that can be exploited in chemosensing devices. The key idea behind using MPc-based heterostructures in sensing devices is to benefit from the organic heterojunction effects in which free charges (e^−^ and h^+^) are accumulated at the interface in accordance with the workfunction difference of the constituents in the heterostructure. Consequently, the interface region of the heterostructure becomes highly conducting, which carrier dynamics can be altered by chemical doping by oxidizing and reducing gases proportionally to their concentrations, which is also the basis of their chemosensing applications.

MPc-based organic heterostructures were used mainly in OFET and MSDI device configurations of which MSDI were used only for the chemosensing applications while a majority of studies on OFET devices were focused on fundamental electrical property investigations. Nonetheless, some of these OFET devices incorporating MPc heterostructures, such as Cu(F_16_Pc)/CuPc, PMMA/CuPc and PTCDI-C8/CuPc, were reported as chemosensors towards gases like NO_2_ and NH_3_. The advantage of using heterostructures was highlighted, which led to achieve larger changes in field-effect mobility and saturation current of the devices under exposure to these gases compared to OFET devices using a homogeneous structure with only one material. Compared to OFET chemosensors, MSDI device configurations have invited larger research interest in MPc heterostructures for chemosensing applications. Many studies, a majority from our group, have reported MSDI sensors incorporating monocyclic and bicyclic MPcs in heterostructures such as Cu(F_16_Pc)/LuPc_2_. These sensing devices displayed very high sensitivities and fast responses and recovery kinetics towards oxidizing gases (NO_2_ and ozone) as well as reducing gases (NH_3_) when used in conductometric or impedimetric transduction modes. Some of these research works focused on understanding the sensor working principles and it was reported that the semiconducting nature of the sublayer determined the n- or p-polarity of the sensor under exposure to redox gases. A survey of recent studies on MSDI devices found that attempts are being made to further improve the sensing performances by incorporating an additional organic-organic interface through electrochemical grafting or electropolymerization of substituted benzenes on the electrode surface. Indeed, by following this approach, NH_3_ sensors displaying high sensitivity, selectivity and stability in response at variable rh were realized having a LOD of 140 ppb, which is among the best NH_3_ sensing performances ever reported.

In the literature surveys presented in this manuscript, the advantages of using MPc-based heterostructures in enhancing gas sensing performances have been highlighted. However, there are still some pertaining issues such as low sensitivity of ambipolar sensors based on MPc heterostructures and the slower kinetics of these sensors compared to their metal oxide counterparts. Ambipolar gas sensors have advantages of dual mode operation (device operating in negative and positive modes), bias-dependent selectivity [79], miniaturized and multiplexed detection platforms and lower fabrication cost compared to their inorganic counterparts. By maximizing the organic heterojunction effects, the charge carriers in the device conduction channel can be suitably enhanced, which can help overcoming the relatively lower sensing response of such devices. Such optimization of charge carriers would require newer designs as well as materials in the organic heterostructure to be used as sensor active layer.

Regarding the newer design of organic heterostructure, creating an additional organic-organic junction by electrochemical grafting is an important and less explored strategy which should be further studied extensively using larger family of benzene molecules substituted with electron donating and accepting groups. To enhance the organic heterojunction effects, new and emerging organic nanomaterials such as graphene or its oxides should be associated with MPc. A few such studies have been reported in recent researches involving CuPc/reduced graphene oxide and CoPc/reduced graphene oxide [80,81]. The NH_3_ sensing properties of chemiresistors based on these hybrid materials have shown improvements, demonstrated by experimental detection of 200 ppb of NH_3_ and long operation stability. Moreover, a recent study on graphene/metal oxide heterostructure have exhibited ultrafast response and recovery kinetics (response and recovery time as 21 s and 41 s) in NH_3_ detection [82]. Thus, graphene-based materials have high potential to develop organic heterostructures to maximize heterojunction effects and apply them in gas sensing.

## Figures and Tables

**Figure 1 sensors-20-04700-f001:**
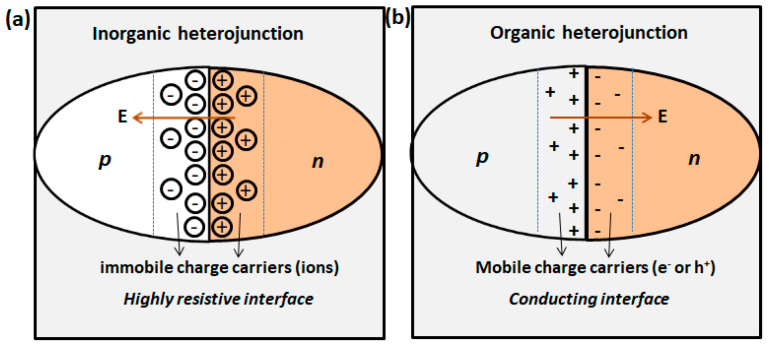
Typical p-n heterojunction formation and interfacial charges alignments in conventional inorganic semiconductor (**a**) and organic semiconductor (**b**). (adapted from [18]). The arrangements of mobile (e^-^ or h^+^) and immobile (ions) charges at the interface and associated direction of electric field are shown.

**Figure 2 sensors-20-04700-f002:**
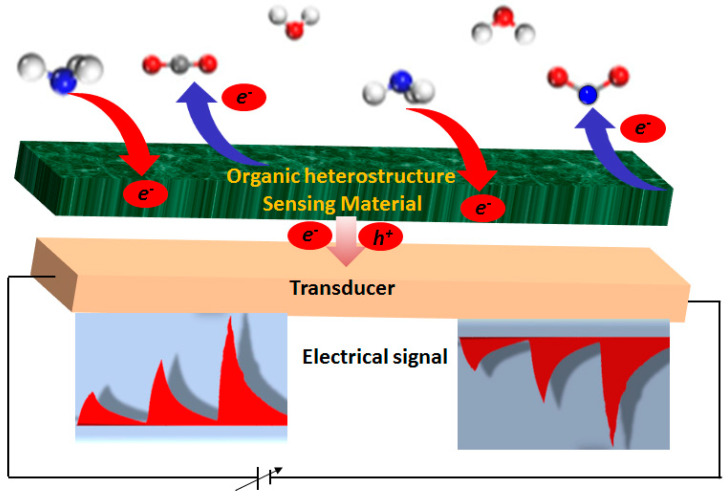
Scheme of the sensing mechanism for redox gases detection using organic heterostructures.

**Figure 3 sensors-20-04700-f003:**
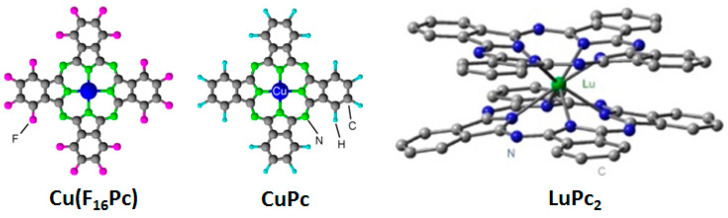
Structures of Cu(F_16_Pc), CuPc and LuPc_2_.

**Figure 4 sensors-20-04700-f004:**
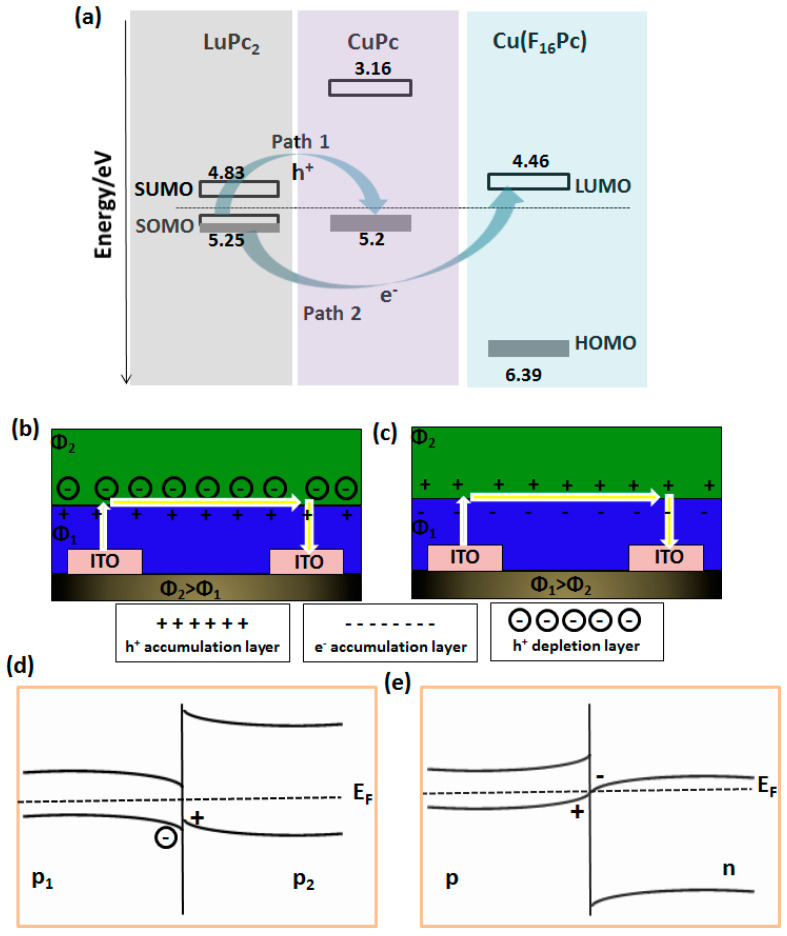
Charge transfer pathways between energy levels at the interface of CuPc/LuPc_2_ (path 1) and Cu(F_16_Pc)/LuPc_2_ (path 2) heterojunction. The energy of frontier molecular orbitals of different MPc are depicted (**a**). The redistribution of charge carriers at the interface and induced band bending are shown, if Φ_top layer_ > Φ_sublayer_ (**b**,**d**) and Φ_top layer_ < Φ_sublayer_ (**c**,**e**). The legends at the bottom of (**b**,**c**) represent the nature of charge accumulation layer at the interface (Adapted from [47,53]).

**Figure 5 sensors-20-04700-f005:**
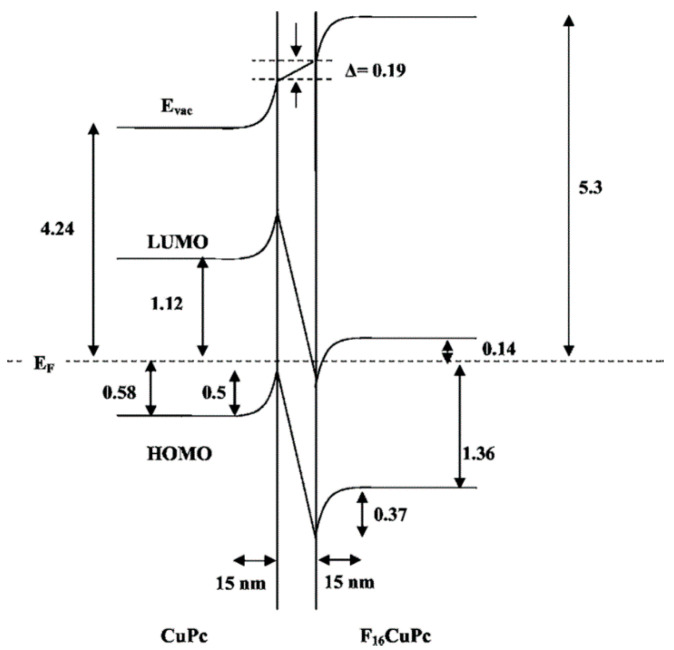
Representation of band bending of HOMO and LUMO levels in Cu(F_16_Pc)/CuPc heterostructure associated with overlapping of HOMO of CuPc and LUMO of Cu(F_16_Pc) [56].

**Figure 6 sensors-20-04700-f006:**
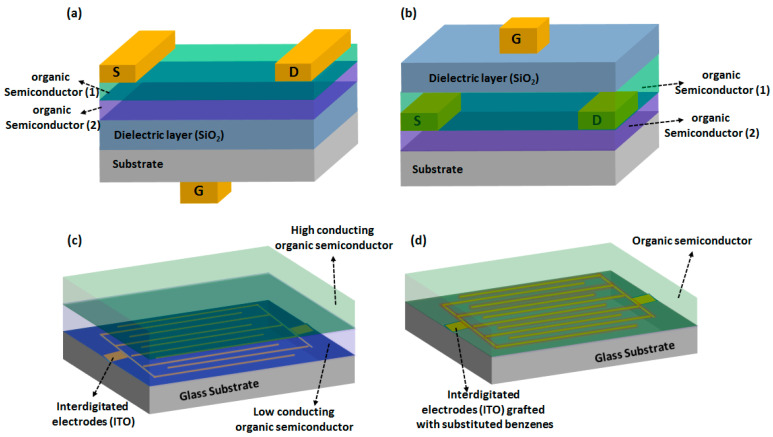
Scheme of OFET device designs in suspended gate (**a**) and top gate (**b**) configuration. Scheme of MSDI (**c**) and double lateral heterojunction (**d**) device.

**Figure 7 sensors-20-04700-f007:**
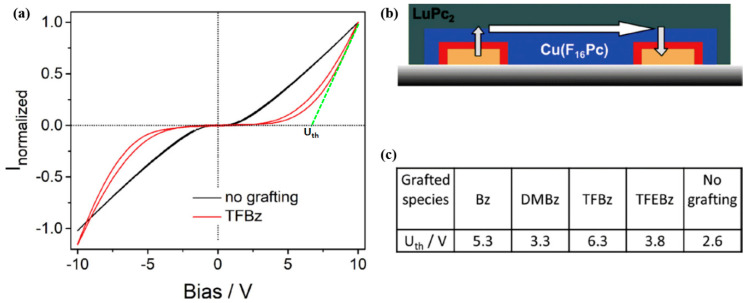
Normalized I(V) curves of a Cu(F_16_Pc)/LuPc_2_ bilayer deposited on electrografted ITO (red) and on unmodified ITO (black) (**a**). Scheme of the device configuration (**b**) and table depicting the apparent energy barrier of the device obtained for electrografting of benzene (Bz), dimethoxy-benzene (DMBz), tetrafluoro-benzene (TFBz) and Trifluoroethoxy benezene (TFEBz) (**c**) [14].

**Figure 8 sensors-20-04700-f008:**
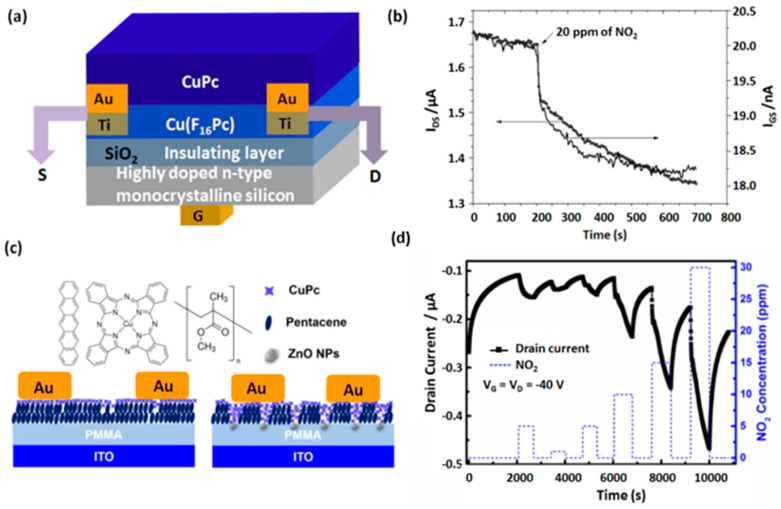
Scheme of OFET design having a suspended gate, Si substrate, SiO_2_ dielectric, Au-Pt as source and drain electrodes and the Cu(F_16_Pc)/CuPc bilayer (**a**) and change in drain and fate current under 20 ppm of NO_2_ exposure (**b**) (adapted from [66]). Scheme of OFET device design, molecular structure of constituents and the microstructure of CuPc/Pentacene heterostructure with ZnO/PMMA dielectric layer on a ITO substrate (**c**). Variations in drain current with 10 min exposure to NO_2_ (in a concentration range of 0.5–15 ppm) and 10 min recovery in dry air (**d**) (adapted from [67]).

**Figure 9 sensors-20-04700-f009:**
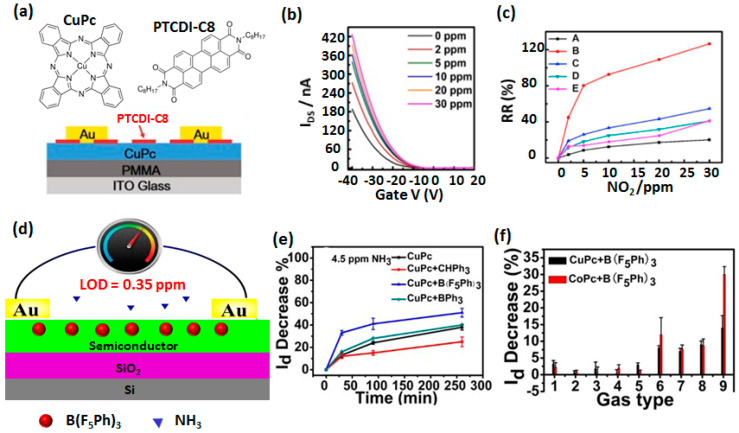
Molecular structure of CuPc and PTCDI-C8 and OFET device design employing CuPc/PTCDI-C8 heterostructure, Au electrode, PMMA dielectric on a ITO substrate (**a**). The variation of drain current at V_DS_ = 40 V under NO_2_ exposure in a range of 0–30 ppm (**b**) and calibration curves depicting variations of RR (calculated from change in saturation current) with NO_2_ concentration for devices having variable thickness of PTCDI-C8 (A: 0 nm, B: 0.5 nm, C: 1 nm, D: 1.5 nm and E: 2 nm) (**c**). (adapted from [68]) Scheme of OFET configuration having a blend of CoPc or CuPc and TPFB on a Si substrate, SiO2 dielectric and Au drain and source electrodes (**d**). The percentage decrease of drain current with time of different heterostructures at NH_3_ exposure of 4.5 ppm (**e**). The percentage decrease of drain current for CuPc+TPFB and CoPc+TPFB heterostructures based OFET upon exposure to different gases vapor (**f**). Numbers on x-axis correspond as follow 1, methanol (2000 ppm); 2, acetone (1800 ppm); 3, dichloromethane (3900 ppm); 4, ethyl acetate (1500 ppm); 5, 5% H_2_ (50000 ppm); 6, isopropylamine (10 ppm); 7, isobutylamine (10 ppm); 8, H_2_S (5 ppm); 9, NH_3_ (4.5 ppm). (adapted from [69]).

**Figure 10 sensors-20-04700-f010:**
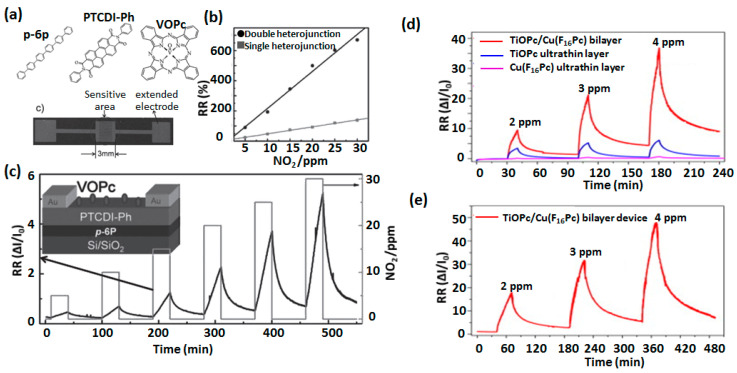
Structure of p-6P, PTCDI-Ph, VOPc and optical image of the device (**a**). Comparison of calibration curves (RR vs NO_2_ concentration) of single (p-6P/PTCDI-Ph) and double (p-6P/PTCDI-Ph/VOPc) heterojunction OFET in NO_2_ concentration range 5–30 ppm (**b**). Response curves of the double heterojunction OFET as a function of time at different alternate exposure of NO_2_ in the range of 5–30 ppm. The configuration of the OFET device is shown in the inset of (**c**) (adapted from [44]) Comparison of response curves as a function of time for TiOPc, Cu(F_16_Pc) and TiOPc/Cu(F_16_Pc) bilayer based OFET device at three different NO_2_ concentrations (**d**) and similar curves of the bilayer OFET at longer exposure and recovery time (**e**). (adapted from [70]).

**Figure 11 sensors-20-04700-f011:**
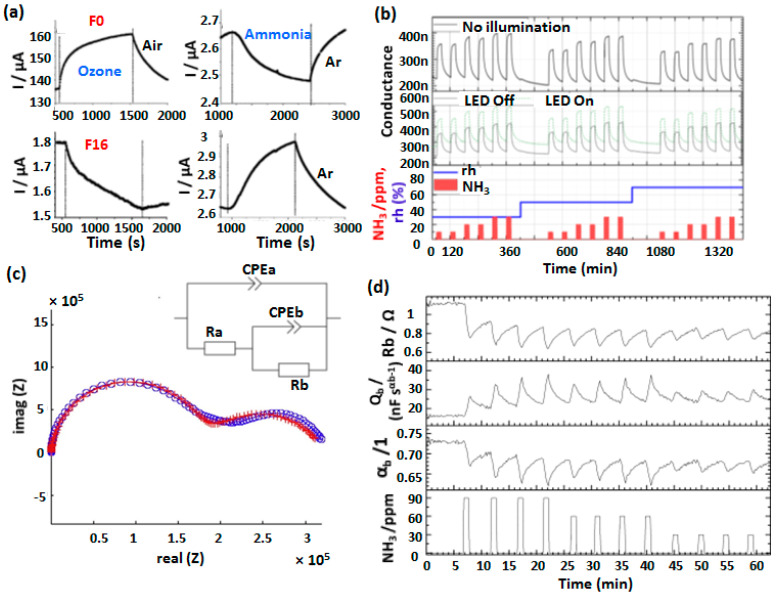
Current variations of CuPc/LuPc_2_ and Cu(F_16_Pc)/LuPc_2_ MSDI under exposure to 90 ppb of ozone and 35 ppm of NH_3_ (**a**). (adapted from [20]) Variations of conductance in dark (upper row) and with light (middle row) at different NH_3_ (red bars) and rh (blue line) levels (**b**) [71]. The Nyquist diagram of Cu(F_16_Pc)/LuPc_2_ MSDI representing variation of real and imaginary impedances (**c**) and the equivalent impedance circuit (inset of c). The variations of interfacial charge transfer coefficient (α) and other impedance circuit parameters as a function of NH_3_ concentrations (**d**) [47].

**Figure 12 sensors-20-04700-f012:**
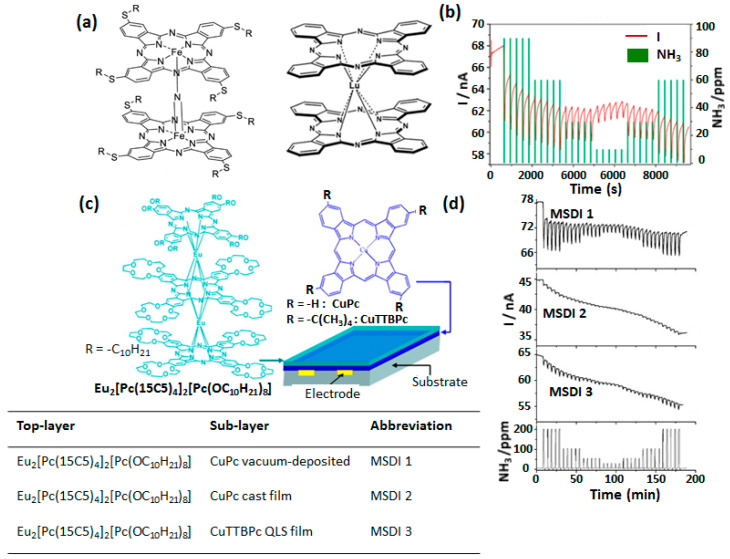
Structure of alkylthio-tetrasubstituted μ-nitrido diiron phthalocyanines and LuPc_2_ (**a**) and NH_3_ sensing performance of related MSDI studied in a range of 10–90 ppm (**b**) [72]. Structure of triple-decker phthalocyanine Eu complex and substituted CuPc and related MSDI device (**c**). Current-time response curves of different MSDI recorded with alternate exposure to NH_3_ in a concentration range 50–200 ppm and recovery under clean air (**d**) [73].

**Figure 13 sensors-20-04700-f013:**
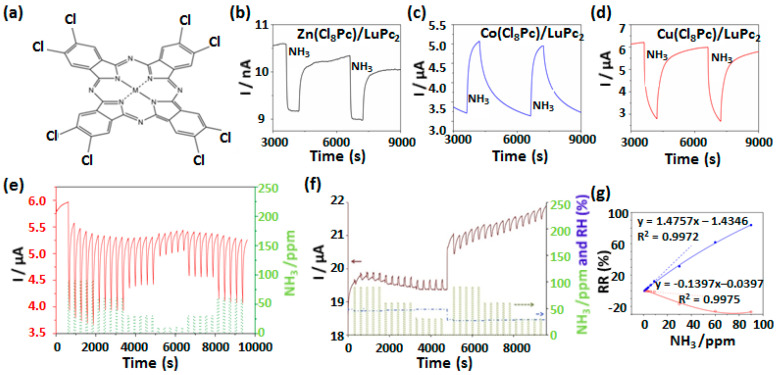
Structure of octachloro-MPc complexes (M: Zn, Co, Cu) (a) and current variations towards alternate exposure to 90 ppm NH_3_ and recovery under clean air for Zn(Cl_8_Pc)/LuPc_2_ (**b**), Co(Cl_8_Pc)/LuPc_2_ (**c**) and Cu(Cl_8_Pc)/LuPc_2_ (**d**). Current variations as a function of time for Cu(Cl_8_Pc)/LuPc_2_ (**e**) and Cu(F_8_Pc)/LuPc_2_ (**f**) devices at different concentrations of NH_3_ in a range of 10–90 ppm. The calibration curves depicting RR variation with NH_3_ concentration are shown for Cu(Cl_8_Pc)/LuPc_2_ and Co(Cl_8_Pc)/LuPc_2_ MSDI (**g**) [53,74].

**Figure 14 sensors-20-04700-f014:**
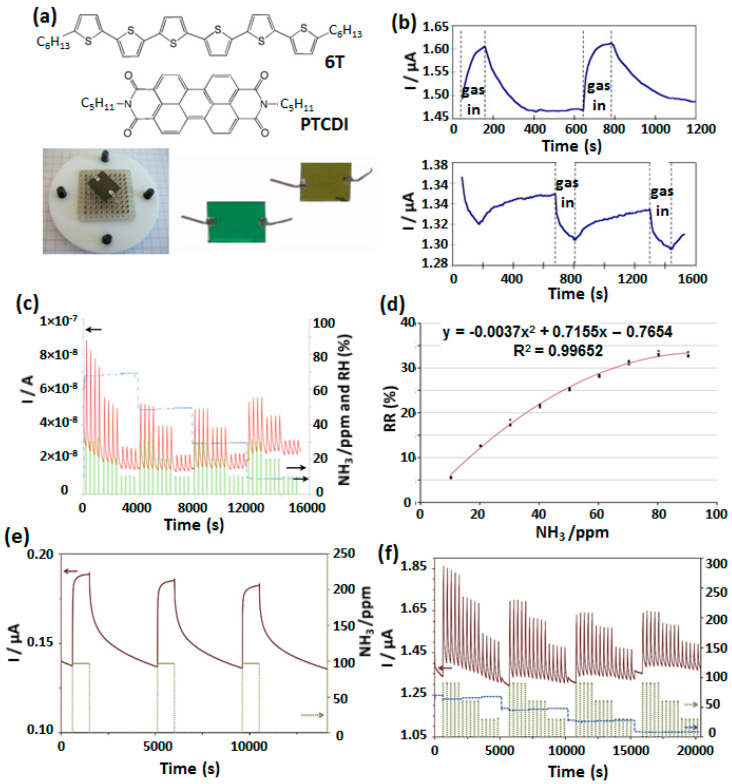
Structure of 5,5′-dihexyl-α,ω-sexithiophene (6T) and the N,N′-dineopentyl-3,4,9,10-perylenetetracarboxylic-diimide (PTCDI) and optical image of the MSDI device (**a**) and its current variation under exposure to 400 ppb ozone (upper curve) and 200 ppm NH_3_ (bottom curve) (**b**) [76]. Current variation of PTCDA-based MSDI at different NH_3_ concentration (10, 20 and 30 ppm) and rh in the range of 30–70% (**c**) and sensor calibration curve depicting change in RR with NH_3_ concentration (**d**) [77]. Current variations of n-MSDI based on TPDO sublayer and LuPc_2_ top layer under 90 ppm NH_3_ exposure for 15 min and recovery under clean air for 1 h (**e**). The sensor response at different NH_3_ concentration (90, 60 30 ppm) and in the rh range of 10–70% (**f**) [78].

**Figure 15 sensors-20-04700-f015:**
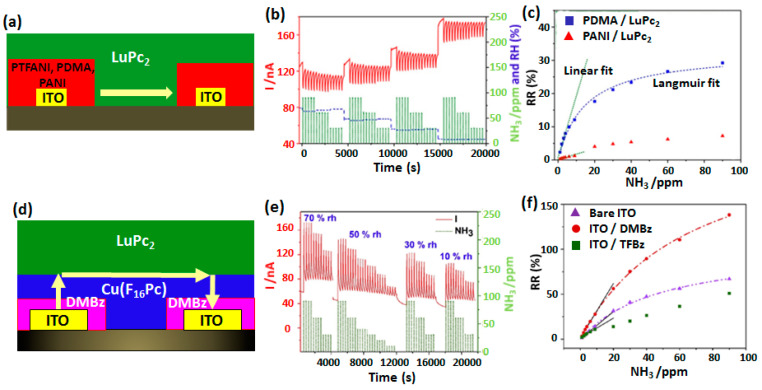
Scheme of Double Lateral Heterojunction of an electropolymerized (PTFANI, PDMA, PANI) organic film on ITO electrodes and PVD coated LuPc_2_ over it (**a**). Change in PTFANI DLH current at different NH_3_ concentrations (10–90 ppm) and rh (10–70%) (**b**). The variation in RR as a function of NH_3_ concentration in a range of 1–90 ppm (**c**) [15,63]. Scheme of DMBz (from DMA precursor) modified Cu(F_16_Pc)/LuPc_2_ MSDI on ITO electrodes (**d**) and its current variation at different NH_3_ concentration (10–90 ppm) and rh (10–70%) (**e**). Calibration curve (RR vs NH_3_ concentration) of different modified MSDI (**f**) [14].

**Table 1 sensors-20-04700-t001:** Comparison of gas sensing properties of different OFET and MSDI devices incorporating MPc-based heterostructures.

Devices	RR [%]	Gas Concentration [ppm]	S [% ppm^−1^]	LOD [ppm]	Range [ppm]	Ref.
Cu(F_16_Pc)/CuPc OFET	18	20 (NO_2_)				[66]
Pentacene/CuPc OFET	193	15 (NO_2_)			0.5–15	[67]
CuPc/PTCDI-C8 OFET	45	2 (NO_2_)			2–30	[68]
CuPc/TPFB OFET	33	4.5 (NH_3_)		0.35	0.45–20	[69]
CoPc/TPFB OFET	37	4.5 (NH_3_)		0.35	0.45–20	[69]
p-6P/PTCDI-ph/VOPc OFET	90	5 (NO_2_)			5–30	[44]
CuPc(COOC_8_H_17_)_8_/CuPc(OC_8_H_17_)_8_ OFET	300	600 (ethanol)	0.49	100	200–1400	[70]
TiOPc/Cu(F_16_Pc) OFET	35	4 (NO_2_)		0.25	1–5	[17]
Co(Cl_8_Pc)/LuPc_2_ MSDI	58	90 (NH_3_)	1.48	0.25	1–9	[53]
Cu(Cl_8_Pc)/LuPc_2_ MSDI	35	90 (NH_3_)			30–90	[53]
Cu(Cl_8_Pc)/LuPc_2_ MSDI	77	30 (NH_3_)			0–30	[53]
N-(ttbFePc)_2_/LuPc_2_ MSDI	11	30 (NH_3_)			10–90	[72]
TFBz/Cu(Cl_8_Pc)/LuPc_2_ MSDI	55	90 (NH_3_)	0.14	1.2	1–9	[14]
Bz/Cu(F_16_Pc)/LuPc_2_ MSDI	67	90 (NH_3_)	1.5	0.28	1–9	[14]
DMBz/Cu(F_16_Pc)/LuPc_2_ MSDI	138	90 (NH_3_)	3	0.14	1–9	[14]
TFEBz/Cu(F_16_Pc)/LuPc_2_ MSDI	50	90 (NH_3_)	1.1	2	1–9	[14]
CuPc/Eu_2_[Pc(15C5)_4_]_2_[Pc(OC_10_H_21_)_8_] MSDI	5	50 (NH_3_)			15–800	[73]
PTCDA/LuPc_2_ MSDI	34	90 (NH_3_)	0.6		10–30	[77]
TPDO/LuPc_2_ MSDI	26	90 (NH_3_)	0.2		30–90	[78]
PTCDI/ LuPc_2_ MSDI	10	0.4 (ozone)				[76]
PTCDI/ LuPc_2_ MSDI	20	100 (NH_3_)			100–800	[76]
PTFA/LuPc_2_ MSDI	14	90 (NH_3_)	1.05	0.45	1–6	[63]
PDMA/LuPc_2_ MSDI	14	90 (NH_3_)	2.23	0.314	1–6	[15]

rh range in all these reported works between 40–50%. All these works report room temperature studies. Empty cells in the table indicates non-availability of the sensing parameters in the literature.

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
