# Peer review of "Organic Heterojunction Devices Based on Phthalocyanines: A New Approach to Gas Chemosensing"

_sensors, 2020, doi:10.3390/s20174700_

Round 1

Reviewer 1 Report

This review is well written and organized. Metal phthalocyanines have wide application as active layers of various electronic devices, including organic heterojunction ones. Therefore it can be attracted for scientists working not only in the field of phthalocyanines but also in the field of chemical sensors. There are just a few comments to make this review even more informative

  1. Please provide examples of recently published reviews on this topic and indicate more clearly what new information this review contains.
  2. Authors report on the advantages of heterojunction  devices in comparison with chemiresistive sensors based on phthalocyanine films towards nitrogen dioxide. Are there any advantages of these devices in relation to ammonia or other reducing gases?
  3. When discussing the workfunction of MPcs (lines 203-209) the appropriate references should be provided.
  4. Fig. 7 (b). The expansion of abbreviation “Bz, DMBz, TFBz, TFEBz” in the table should be given.
  5. Line 349. The expansion of abbreviation of “RR values” should be provided.
  6. Quality of Figs. 9(d), 10(a) should be improved. Some of the inscriptions in these figures are difficult to read.

Reviewer 2 Report

See the attached Reviewer's comment file

Reviewer 3 Report

Organic Heterojunction Devices Based on  Phthalocyanines: A New Approach to Gas  Chemosensing

Abhishek Kumar *, Rita Meunier-Prest and Marcel Bouvet *

The abstract clearly introduces the theme of the review. The sensors are selective for the NO2, ozone and NH3?

The introduction explains in a simple and clear way the advantage of the creation of a heterojunction in organic semiconductor. The authors clearly demonstrate the different ways of creating organic heterojunctions and the advantages/disadvantages. Also, they explain why phthalocyanines are one of the most studied molecular semiconductors in chemosensing. In the end of the introduction the authors point the different aspects that will be discussed in the review, helping the reader to guide the reading.

  1. Suitability of phthalocyanines in heterostructure based gas sensors

This part is written in a quite confusing way. A non-area reader will not understand the different parameters referred. I suggest the authors re-written this part.

2.2. Metal phthalocyanines in organic heterostructures

In this part I would like to see more detail about the metollophthalocyanines. For instance, a discussion and comparation between different metals: which are the best ones and why.

Pag 6. Line 206“Moreover, the  workfunction of MPc can also be tuned by the substituents electronic effects, which can be finely  engineered to align with the electrode (such as gold or ITO) workfunctions for an efficient charge  injection and reception during sensors operation. Indeed, the workfunctions of some of the  commonly used electroactive”

The authors should detail more which the effect of the substituents in the workfunction of MPc.  

3.2. MPc heterostructure integration in chemosensing devices

The authors say that are mainly two device structures incorporating MPc based heterostructure for chemosensing applications, but 90% of this part only MSDI is referred. Also, more detail about OFET should be given. Since in the next sections the authors will discuss both types I am not convinced of the need of this section. 

4.1 The authors present some examples. I would like to see an opinion about the selected examples. The literature revision is performed since which year?

4.2 The table presented in this section compares both devices. It helps the readers to make some conclusions.

In my opinion the authors performed a good conclusion about the revision performed including some aspects for the future. Although it is not clear which is the interval of literature research and if these are the only examples in literature. If not, why the authors selected this examples?

I think that this review as it is not ready for publication.

Round 2

Reviewer 3 Report

The authors made all the suggested changes. The article is well written and in terms of organization and clarification of the parameters it is more noticeable. In this way, I believe that the review article is ready to be published.